# Baseline IgM Amounts Can Identify Patients with Poor Outcomes: Results from a Real-Life Single-Center Study on Classical Hodgkin Lymphoma

**DOI:** 10.3390/cancers16040826

**Published:** 2024-02-18

**Authors:** Andrea Duminuco, Gabriella Santuccio, Annalisa Chiarenza, Amalia Figuera, Giovanna Motta, Anastasia Laura Caruso, Alessandro Petronaci, Massimo Ippolito, Claudio Cerchione, Francesco Di Raimondo, Alessandra Romano

**Affiliations:** 1Hematology Unit with BMT, A.O.U. Policlinico “G.Rodolico-San Marco”, 95123 Catania, Italy; santuccio.gabriella@gmail.com (G.S.); annalisa.chiarenza@gmail.com (A.C.); amaliafiguera@gmail.com (A.F.); gmotta72@gmail.com (G.M.); laura75.caruso@libero.it (A.L.C.); alessandropetronaci@outlook.it (A.P.); diraimon@unict.it (F.D.R.); 2Nuclear Medicine Center, Azienda Ospedaliera Cannizzaro, 95021 Catania, Italy; ippolitomas@yahoo.it; 3Hematology Unit, IRCCS Istituto Romagnolo per lo Studio dei Tumori (IRST) “Dino Amadori”, 47014 Meldola, Italy; claudio.cerchione@irst.emr.it; 4Department of General Surgery and Medical-Surgical Specialties, Hematology Section, University of Catania, 95123 Catania, Italy

**Keywords:** IgM, score at diagnosis, prognosis, IPS, Hodgkin lymphoma, PET-2

## Abstract

**Simple Summary:**

Hodgkin Lymphoma (HL) has a prognostic assessment often relying on PET scans after two ABVD treatment cycles (PET-2). This retrospective, observational, single–center, real-life study, evaluated 212 newly diagnosed HL patients, highlighting the role of IgM levels at diagnosis. IgM ≤ 50 mg/dL at baseline emerges as an independent predictor for progression-free survival in classic HL, offering an early and crucial prognostic marker. Additionally, the combined factors of IgM ≤ 50 mg/dL and the presence of a large nodal mass (<7 cm) serve as predictors of outcomes in classic HL, emphasizing the importance of these parameters in understanding and managing the disease above all in cases with unclear PET-2 and guaranteeing the best of care for the patients with an improved outcome.

**Abstract:**

Hodgkin Lymphoma (HL) is characterized by an inflammatory background in which the reactive myeloid cells may exert an immune-suppressive effect related to the progression of the disease. Immunoglobulin M is the first antibody isotype produced during an immune response, which also plays an immunoregulatory role. Therefore, we investigated if, as a surrogate of defective B cell function, it could have any clinical impact on prognosis. In this retrospective, observational, single–center study, we evaluated 212 newly diagnosed HL patients, including 132 advanced-stage. A 50 mg/dL level of IgM at baseline resulted in 84.1% sensitivity and 45.5% specificity for predicting a complete response in the whole cohort (area under curve (AUC) = 0.62, *p* = 0.013). In multivariate analysis, baseline IgM ≤ 50 mg/dL and the presence of a large nodal mass (<7 cm) were independent variables able to predict the clinical outcome, while, after two cycles of treatment, IgM ≤ 50 mg/dL at baseline and PET-2 status were independent predictors of PFS. The amount of IgM at diagnosis is a valuable prognostic factor much earlier than PET-2, and it can also provide information for PET-2-negative patients. This can help to identify different HL classes at risk of treatment failure at baseline.

## 1. Introduction

Hodgkin lymphoma is a B cell malignant neoplasm that is distinguished from all other tumors by its peculiar anatomopathological features [1].

The Reed-Sternberg neoplastic cells are surrounded by an overwhelming inflammatory infiltrate, which plays a key role in the development and progression of the disease [2].

Currently, 18F-Fluoro-2-Deoxy-Glucose Positron Emission Tomography (FDG-PET), performed early after the first two cycles of chemotherapy (PET-2), has been established as the main prognostic factor able to predict clinical outcome and treatment failure [3,4].

A positive PET2 status, as a surrogate of the tumor’s low chemo-sensitivity, reflects the persistence of inflammatory cells with high glycolytic activity [5,6], and it is considered the most valuable tool to set up a risk-adapted strategy [7,8].

However, the negative predictive value of interim-PET is not absolute since about 10% of patients who tested negative failed to achieve a continued complete response.

To improve this risk assessment and avoid therapeutic failures, it would be useful, in clinical management, to identify treatment-independent prognostic factors already at diagnosis and not just after starting treatment.

For this purpose, novel insights in HL biology have been indicated as possible prognostic factors [9], although most of them have not yet been validated in prospective clinical trials.

Until now, most studies carried out have tried to shed light mainly on the controversial role of acquired cell-mediated immunity [10,11,12]; in the meantime, an increasing amount of evidence in Hodgkin Lymphoma suggests that the clonal expansion of altered myeloid progenitors with immunosuppressive activity, known as myeloid-derived suppressed cells (MDSCs) [13] could be the basis for the impairment of the myeloid axis in HL. Previous reports show that granulocytes are dysfunctional in HL, and high levels of immature MDSCs in peripheral blood have a negative predictive value on the patient’s outcome, independently by adopting a therapeutic strategy based on PET-scan results [14,15].

Moreover, an increase in the absolute neutrophil count (ANC) is a frequent finding in HL with a negative prognostic significance [16], although it is not included in the International Prognostic System (IPS) [17]. A high neutrophil-to-lymphocyte ratio (NLR) and a low lymphocyte-to-monocyte ratio (LMR) have been reported as negative prognostic factors by several retrospective studies [18,19], and a more recent report has tested their predictive values in relation to the long-term outcomes of patients treated with a PET-2 risk-adapted strategy [20]; however, the available data in the PET-2 era is still limited, and further efforts should be made in this direction.

On the other hand, the acquired humoral immunity still appears to be almost unexplored.

The origin of the Reed-Sternberg cell has been demonstrated to be from a proliferative B cell of the germinal center, which has acquired some genetic mutations that are able to prevent apoptosis; the proof of this derivation has been provided by the presence of clonal-type rearrangements in the coding genic region for the heavy chains of immunoglobulins, which physiologically are involved in the effector phase of humoral immunity [21].

Starting from this evidence, we investigated if a low amount of immunoglobulin M at baseline, as a surrogate of defective B cell function, could have any clinical impact on prognosis.

Although never tested in the field, baseline values of immunoglobulins could be included among those clinical parameters that, from the beginning, can predict the clinical course of the disease in newly diagnosed HL patients.

## 2. Patients and Methods

### 2.1. Patient Selection

In this retrospective, observational, single-center study, we evaluated 212 newly diagnosed HL patients who accessed the Hematology Division of the University of Catania consecutively. The HL diagnosis was the only criterion needed to include patients in the study, regardless of age and the histological subtype. Exclusion criteria were known active infective diseases (including HIV and tuberculosis infection), the presence of autoimmune diseases, the use of oral or parenteral corticosteroids, or other immunosuppressant therapy less than 3 months before diagnosis.

All patients provided their written informed consent in accordance with the Declaration of Helsinki.

For all patients, the diagnosis was made via histological evaluation, following an excisional biopsy of the involved lymph node; an immunophenotype study was performed for histological classification. Bone marrow aspiration was performed to evaluate the disease extent and define the correct staging. Before starting treatment, a blood sample was collected for a complete blood count (CBC) evaluation and routine biochemical examinations.

White blood cell (WBC) counts and types (neutrophils, eosinophils, basophils, lymphocytes, and monocytes) were determined using an electrical impedance method in an automatic blood counter device. The quantification of immunoglobulins (IgG, IgA, IgM) at baseline was performed via nephelometry, as part of standard initial clinical work-up. The baseline characteristics of patients included in the study are shown in Table 1.

### 2.2. Staging, Risk Assessment, and Treatment

For each patient, we assessed the stage of disease at diagnosis according to the Ann-Arbor classification system, classifying patients as affected by an early (stage I–II) or advanced-stage disease (stage IIB, III–IV). The risk assessment was carried out by calculating the International Prognostic Score (IPS). Functional and morphological evaluation of the patient at baseline was obtained both with Positron Emission Tomography (PET) and Computed tomography (CT) scans.

For patients with advanced disease (Ann-Arbor stage ≥IIB), treatment consisted of six cycles according to the ABVD regimen (Doxorubicin, Bleomycin, Vinblastine, and Dacarbazine), followed by involved field radiotherapy, if clinically indicated. Positive PET-2 patients were shifted to a BEACOPP regimen (Bleomycin, Etoposide, Doxorubicin, Cyclophosphamide, Vincristine, Procarbazine, and Prednisone) for eight cycles and, in the case of persistent disease, autologous stem cell transplant was performed. Early-stage disease patients received two, four, or six cycles of ABVD chemotherapy plus involved field radiotherapy, as clinically indicated.

Both baseline PET and PET-2 were performed using a standard technique, and a semi-quantitative Deauville score was attributed to each image. A minimally positive scan was defined as any scan with any residual FDG uptake outside the physiological areas of tracer concentration [22].

### 2.3. Statistical Analysis

Qualitative results were summarized as counts and percentages. Descriptive statistics were generated for analyzing the results, and a *p*-value under 0.05 was considered significant.

The primary endpoint of the study was to establish if baseline immunoglobulins could have any impact on the progression-free survival (PFS) of newly diagnosed HL patients. The secondary endpoint included the evaluation of baseline immunoglobulins to predict overall survival (OS) or PET-2 status in the whole cohort or only in the cohort of advanced-stage patients.

The correlation of baseline IgM with other established prognostic factors in HL was assessed with Pearson’s chi-square test (or Fisher’s Exact Test) for categorical parameters and with a Mann–Whitney or Wilcoxon signed rank test for continuous parameters.

Progression-free survival (PFS) was calculated from the time of inclusion of patients in the study until the date of progression, relapse, or death or until the date at which the patient was last known to be in remission. The positivity of PET-2 was not considered an event. PFS was analyzed using the Kaplan–Meier Test. Standard errors were calculated using the Greenwood method, and the 95% confidence intervals were computed as 1.96 times the standard error in each direction.

The Cox Proportional Hazards Model was used to evaluate IgM at diagnosis as a prognostic marker for PFS and to assess and adjust with other known prognostic factors.

All calculations were performed using MedCalc Statistical Software version 19.1.6 (MedCalc Software Ltd., Ostend, Belgium; https://www.medcalc.org; accessed on 10 October 2023) and Addinsoft (2024). XLSTAT statistical and data analysis solution, New York, NY, USA, https://www.xlstat.com/en (accessed on 10 October 2023), was used.

## 3. Results

### 3.1. Baseline IgM Is Reduced in cHL

In this series of 212 consecutive cHL patients, the median age was 31.8 (range, 14.8–76.9), and half of the patients were males (Table 1).

The median ANC, ALC, and NLR at diagnosis were 7.24 × 103/μL, 1.46 × 103/μL, and 4.79, respectively. The median baseline values for IgM, IgA, and IgG were 103.00 (interquartile range (IQR), 50.00–336.00), 211.00 (IQR, 36.00–690.00), and 1169.00 (IQR, 312.00–2763.0) mg/dL, respectively. Low levels of IgM (≤50 mg/dL) and IgG (≤1000 mg/dL) were not associated with age, gender, advanced stage, high WBC, low ALC counts, high IPS, positive PET-2, or increased LDH (Table 2).

### 3.2. Low Baseline IgM Concentrations Are Associated with Clinical Outcomes

After two ABVD cycles, out of 212 patients included in the study, 33 (16%) had positive PET-2 (17 with a Deauville score of 4 and 16 with a Deauville score of 5) and 179 (84%) had a negative PET-2 (Figure 1).

Advanced-stage patients with positive PET-2 (28/132) were allocated to an escalated BEACOPP program (*n* = 18, not completed for toxicity in five cases), or they completed the program with a total of six cycles outside of clinical trials (*n* = 10), followed by high-dose chemotherapy (IGEV or DHAP) and autologous stem cell transplantation; 104 patients with negative PET-2 continued with ABVD for an additional four cycles, followed by consolidation radiotherapy on the initial bulky nodal site of disease in 24 cases (Figure 1). One early death due to infection occurred among advanced-stage patients with negative PET-2. A third PET evaluation was performed at the end of planned treatment (PET-3), and a treatment failure within 2 years was registered in 7/104 patients with negative PET-2 as compared to 12/28 with positive PET-2 (Figure 1).

In 80 early-stage patients, therapy was not modified according to the PET-2 results. Based on disease extension, ABVD cycles were followed by involved field 30 Gy radiotherapy (ifRT), except in four cases where treatment for medical conditions and the patient’s will was extended to six cycles of ABVD. Two of them did not achieve a complete response and were treated with salvage-regimen IGEV and autologous stem cell transplantation (Figure 1). All five patients with positive PET2 progressed within 2 years and received high-dose chemotherapy (IGEV or DHAP) and autologous stem cell transplantation.

After a median follow-up of 70.2 months (range 4.8–214.0 months), 188 patients (88.7%) were in continued complete remission (cCR), 6 patients (3.0%) progressed during therapy or immediately after (during the first six months), and 18 patients relapsed (8.4%) within two years from the end of treatment (Table 1). Among 132 advanced-stage patients, 28 (27.7%) had a positive PET-2 scan. Among 80 early-stage patients, 5 (6%) had a positive PET-2 scan, as summarized in Figure 1. As expected, a positive PET-2 scan was associated with an inferior PFS, as shown in Figure 2.

In addition, 49/212 (23.1%) patients had a baseline IgM value lower than 50 mg/dL (Table 1). Achieving and maintaining a complete response (cCR) was associated with lower median IgM than treatment failure (respectively, 90 versus 48 mg/dL, *p* = 0.0015). A Receiver Operating Characteristic (ROC) Curve analysis identified a level of IgM of 50 mg/dL to discriminate patients with 84.1% sensitivity and 45.5% specificity (area under the curve (AUC), 0.624; *p* = 0.013, Figure 3), while IgG and IgA amounts at baseline failed to identify patients in complete remission (AUC, respectively, 0.55; *p* = 0.28 and 0.52, *p* = 0.61).

Based on ROC analysis, a cut-off level of 50 mg/dL of IgM at baseline was chosen to predict clinical outcomes at 36 or 60 months in a further analysis. Patient allocation based on the clinical outcome and basal amount of IgM is shown in Figure 4.

PFS at 60 months was 54.1% versus 81.1% in patients with IgM ≤ 50 mg/dL or IgM > 50 mg/dL, respectively (*p* < 0.001, Figure 5).

A white blood cell (WBC) count ≥ 15,000 cells/μL (*p* = 0.005), IPS ≥ 3 (*p* = 0.01), large nodal mass > 7 cm (*p* = 0.02), extranodal disease (*p* = 0.03), positive PET-2 (*p* < 0.0001), NLR ≥ 6 (*p* = 0.008), and IgM ≤ 50 mg/dL (*p* < 0.0001) were predictors of 5-year progression-free survival (PFS) in the univariate analysis (Table 3).

To overcome statistical bias due to the retrospective analysis in a limited series, we split randomly, at 50:50, the cohort into a training set (*n* = 106) and a validation set (*n* = 106, Appendix A) to test the contribution of IgM in the risk stratification. In the training set, the 5-year PFS estimates were 54.9% for patients carrying low versus 72.5% for patients carrying normal IgM at baseline (*p* = 0.02, as shown in Appendix A), which was confirmed in the validation set (53.8 vs. 86.6%, Appendix A). Univariate analysis of PFS in training and validation further confirmed the positive PET-2 (*p* < 0.001 in both settings, shown in Appendix A) and IgM ≤ 50 mg/dL (respectively, *p* = 0.02 and *p* < 0.001, in the training and validation set, Appendix A) as predictors of 5-year PFS.

### 3.3. Baseline IgM Can Predict Clinical Outcomes in Advanced-Stage HL Patients

Advanced-stage patients with positive PET-2 had an inferior outcome, with a 5-year PFS of 25.2% versus 80.5% for negative PET-2 patients (*p* < 0.0001). After two cycles of treatment, patients with negative PET-2 carrying IgM ≤ 50 mg/dL had poor outcomes, with a PFS at 60 months of 68.0% versus 84.7% for cases with IgM > 50 mg/dL, *p* = 0.0068 (Figure 6), the only predictor of the outcome in the univariate analysis (Table 4).

Thus, in advanced-stage patients, WBC count ≥ 15,000 cells/μL (*p* = 0.012), IPS ≥ 3 (*p* = 0.01), large nodal mass > 7 cm (*p* = 0.02), extranodal disease (*p* = 0.04), positive PET-2 (*p* < 0.0001), NLR ≥ 6 (*p* = 0.002), and IgM ≤ 50 mg/dL (*p* = 0.0002) were predictors of 5-year PFS in the univariate analysis (Table 4).

In the multivariate analysis, among parameters available at baseline, only IgM title ≤ 50 mg/dL and a large nodal mass > 7 cm were independent predictors of treatment failure, while when including the PET-2 status, only IgM ≤ 50 mg/dL retained its independent prognostic relevance (Table 5).

### 3.4. Combining Baseline IgM and the Presence of Large Nodal Mass Can Predict Clinical Outcomes in Advanced-Stage HL Patients Independently of the PET-2 Status

Based on previous findings, we stratified advanced-stage patients into three groups based on clinical variables available at diagnosis: the low-risk group was defined as the absence of a large nodal mass (LNM) and baseline IgM >50 mg/dL (*n* = 63, 48%); the standard-risk group by either the presence of an LNM or baseline IgM ≤ 50 mg/dL (*n* = 59, 45%); the high-risk group by the concurrent presence of an LNM and IgM ≤ 50 mg/dL (*n* = 10, 7%).

The PFSs at 60 months were significantly different among the three risk groups, at 83.5%, 59.5%, and 40.0%, respectively, *p* < 0.0001 (Figure 7).

## 4. Discussion

This work illustrates the results of a study aimed at demonstrating, for the first time in the field, the pathogenetic involvement of a reduction in immunoglobulins at baseline in clinical outcomes of newly diagnosed HL patients.

In the era of risk-adapted therapy in HL, an appropriate risk classification is needed in order to identify those patients for whom aggressive treatment is required.

The prognostic model based on the International Prognostic System (IPS) has insubstantial clinical utility because only 19% of patients with scores of 4 and 5 had a probability of 7-year-progression-free survival (PFS) < 50% [7].

Currently, PET-2 is the most valid tool for predicting poor outcomes and setting a risk-adapted strategy for treatment, and PET-2-derived information is currently used in clinical practice to switch to intensified or de-escalate treatment [8,23].

However, some issues are still open: PET-2 evaluation provides quite late information because it is available only during treatment when it is likely to believe that the biological mechanisms leading to chemoresistance have already been activated. Moreover, its negative predictive value is not absolute, as about 10% of patients with a negative PET-2 are still at risk of treatment failure [8,23].

For these reasons, the search for prognostic factors able to identify high-risk patients at diagnosis and not during treatment remains an objective that is not fully achieved yet.

From this perspective, several biomarkers have been proposed during the last few years as putative predictors of clinical outcomes, although some of them could not be included because of a lack of reproducibility [12,19,24,25]. Recently, pretreatment and on-treatment circulating tumor DNA (ctDNA) levels have shown a potential role in the dynamic monitoring of classic HL, supporting the utility of noninvasive strategies [26].

Thymus and activation-regulated chemokines (TARCs) and Arginase-1 (Arg1) stand out among the soluble factors that reflect the immune system dysregulation and which can be easily tested at diagnosis with a peripheral blood sample [27,28,29]. These factors are associated with therapeutic success and negativity of PET-2 [30], but their predictive role has not yet been demonstrated in a prospective series of patients treated upfront with a risk-adapted strategy.

The role played by the acquired cell-mediated immunity (especially the regulatory function of T cells) in the development and progression of disease has been largely investigated by an increasing number of studies. Instead, as the side of acquired humoral immunity has been much less explored, one might ask if B cell aggregates can play an active role in intratumoral immunity or if they are mere bystanders in the HL microenvironment. Indeed, so-called immunoparesis, a condition that describes the suppression of uninvolved immunoglobulins, has been recently reported as a precursor marker of the absence of an immune response to the SARS-CoV-2 vaccine in lymphoproliferative disorders [31,32].

In view of these results and because of the B cell derivation of HL, we supposed that reduced levels of immunoglobulins at diagnosis, as a marker of humoral immunity impairment, could contribute to immunological dysfunction, which appears to be the main pathogenetic issue and which has important effects on the clinical outcomes of these patients.

Indeed, in recent years, evidence has emerged to suggest that humoral immunity may contribute to the immune response in cHL, and several immunohistochemical studies have been carried out to assess the immunophenotypic characteristics of these B cell infiltrates; particularly, increased numbers of CD20+ B cells in the tumor infiltrate were shown to favor a good prognosis regardless of the disease stage. Therefore, these results might suggest an active role of B cells in the anti-tumor immune response [33,34].

However, this observation appears to be controversial, as targeting B cells with Rituximab led to a better outcome in cHL patients; moreover, B cells may trigger Hodgkin and Reed-Sternberg cell (HRS) growth through the expression of CD30L and CD40L [35].

Interestingly, high numbers of CD38+ B-cells (including transitional B cells, regulatory B cells, memory B cells, and post-germinal center cells) are associated with worse survival outcomes [34]. In this regard, CD38 signaling can decrease the differentiation of IgM-producing plasma cells [36], a specific cell subset for which a positive effect on the prognosis of these patients has been shown [34].

In our study, we evaluated the survival analyses that have been carried out for the whole cohort of patients, as well as, subsequently, only for patients with the advanced stage of disease, for whom we expected a worse outcome.

Our findings show that the IgM amount at diagnosis is a valuable prognostic factor much earlier than PET-2 and can also provide information for negative PET-2 patients. Therefore, it could be added to known predictive factors in order to identify different HL classes for the risk of treatment failure at baseline.

## 5. Conclusions

In HL, the IgM level at the time of diagnosis could serve as a valuable prognostic indicator before the conclusion of the first two chemotherapy cycles, even offering insights into PET-2-negative patients. This early assessment proves to be beneficial for identifying distinct classes that may be at risk of treatment failure from the outset.

## Figures and Tables

**Figure 1 cancers-16-00826-f001:**
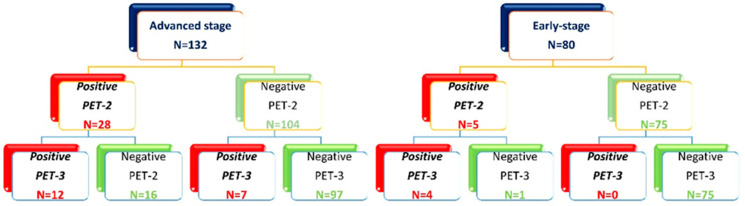
Allocation of patients based on the PET-2 and PET-3 scan.

**Figure 2 cancers-16-00826-f002:**
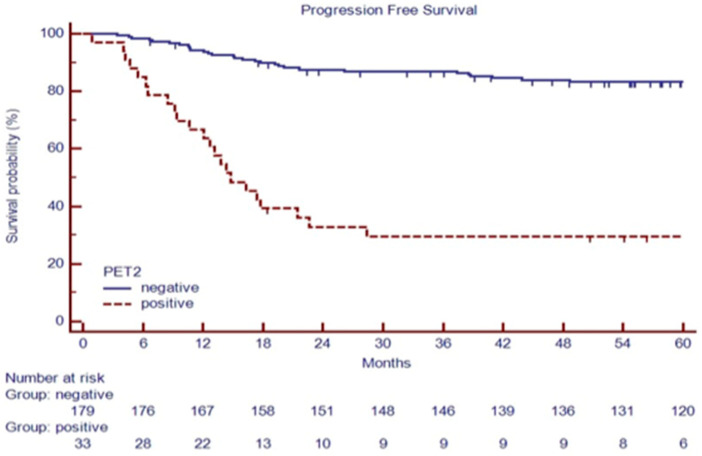
PFS based on PET-2 status in 212 consecutive cHL patients.

**Figure 3 cancers-16-00826-f003:**
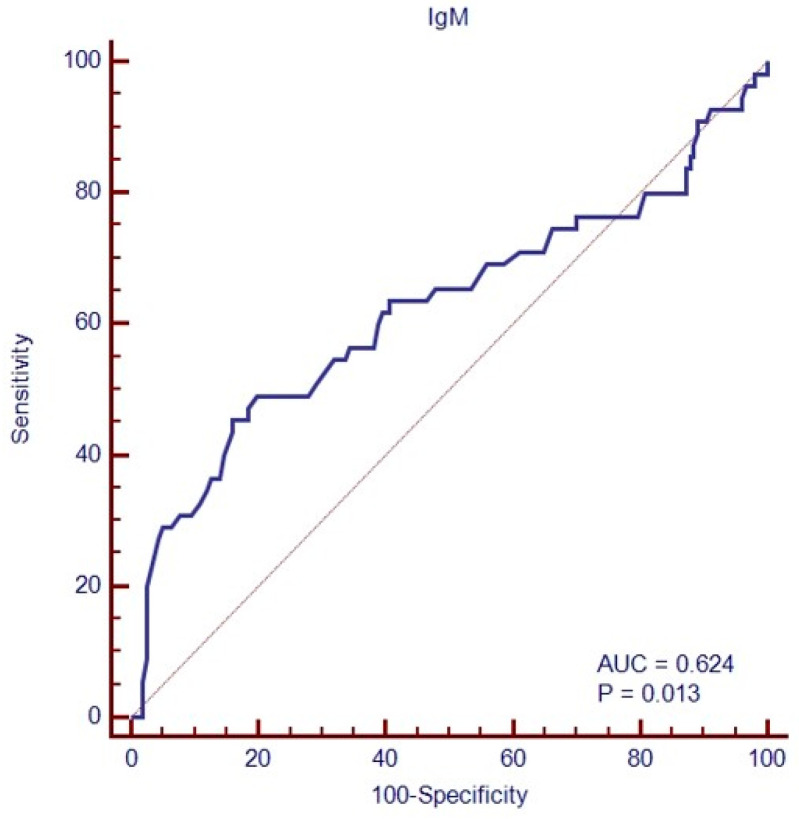
ROC analysis of baseline immunoglobulin (IgM class) amounts to predict treatment failure.

**Figure 4 cancers-16-00826-f004:**
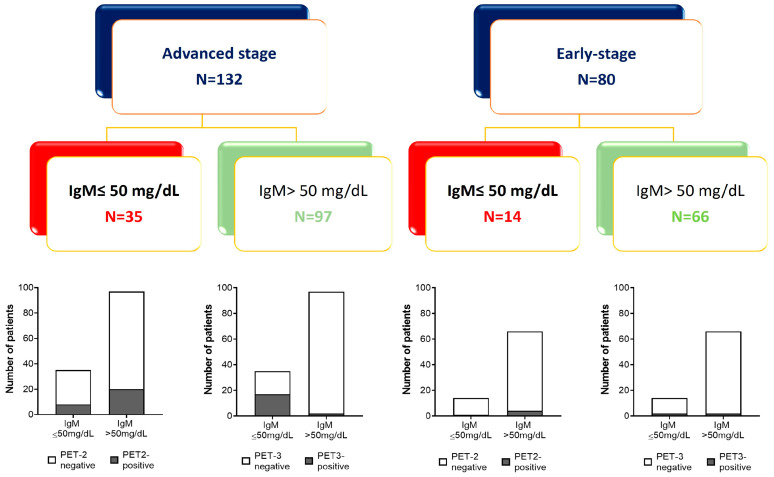
Patient allocation based on the amount of baseline IgM and clinical outcome (PET2 and PET3 scans).

**Figure 5 cancers-16-00826-f005:**
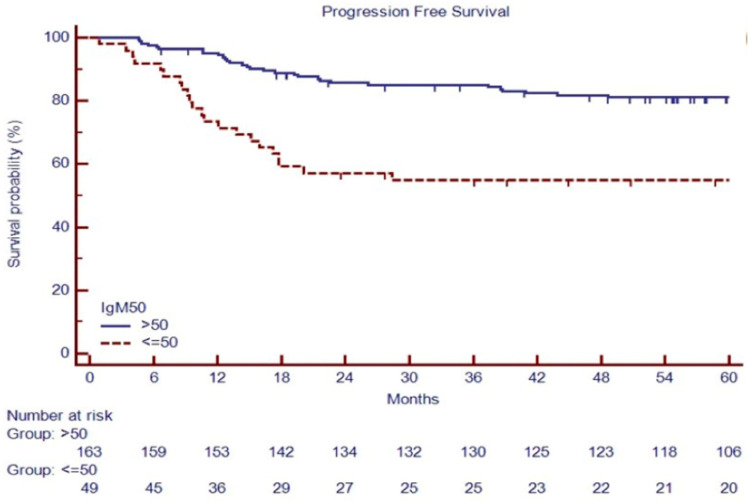
PFS based on baseline IgM in 212 consecutive cHL patients.

**Figure 6 cancers-16-00826-f006:**
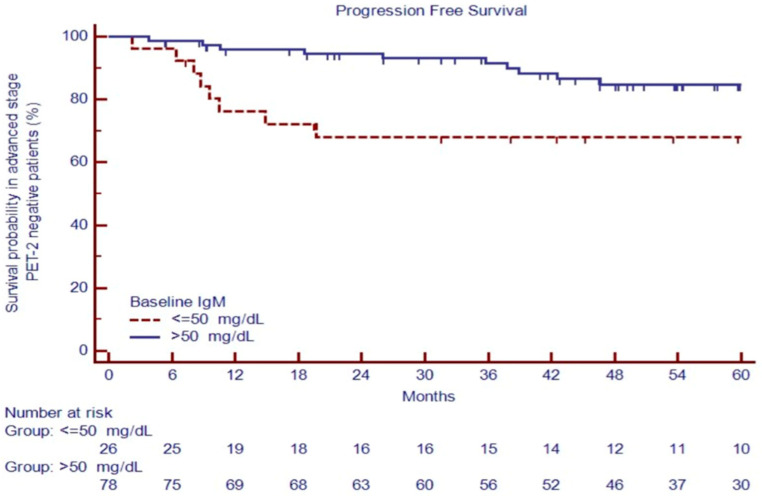
PFS in PET-2 negative patients based on IgM evaluation at diagnosis.

**Figure 7 cancers-16-00826-f007:**
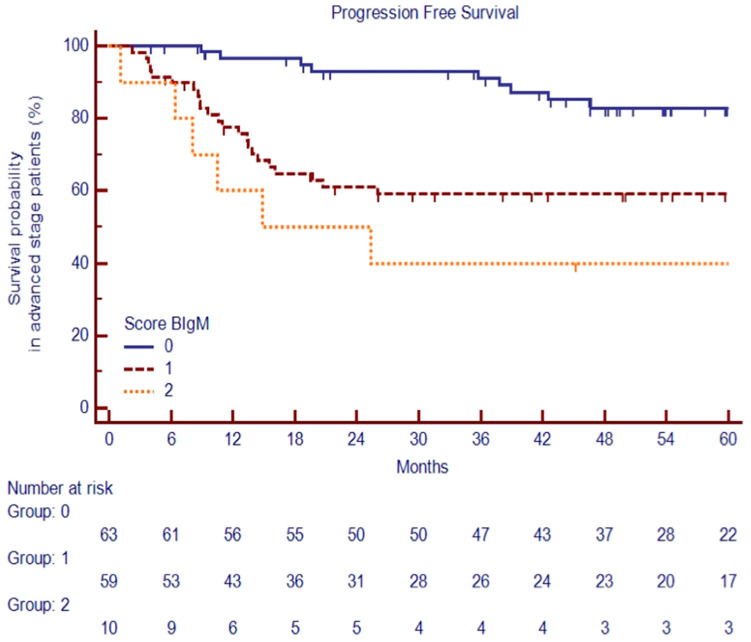
PFS at 60 months among the three risk groups of advanced-stage cHL patients.

**Table 1 cancers-16-00826-t001:** Baseline characteristics of patients included in the study (*n* = 212).

	All Patients (*n* = 212)	All Patients (*n* = 212)			Advanced-Stage Patients		
		IgM ≤ 50, *n* = 49	IgM > 50, *n* = 163	*p*	IgM ≤ 50, *n* = 35	IgM > 50, *n* = 97	*p*
**Age (years)**							
Median (range)	31.8 (14.8–76.9)	31.8 (15.6–76.1)	31.9 (14.9–76.9)	0.89	31.8 (16.0–76.1)	32.7 (14.8–76.9)	0.92
<45, *n* (%)	148 (69.8%)	34 (69.4%)	114 (69.9%)	0.98	24 (68.6%)	66 (68.0%)	0.98
≥45, *n* (%)	64 (30.1%)	15 (30.6%)	49 (30.1%)		11 (31.4%)	31 (32%)	
**Sex, *n* (%)**							
Female	103 (48.6%)	22 (44.5%)	81 (49.7%)	0.62	16 (45.7%)	44 (45.4%)	0.98
Male	109 (51.4%)	27 (55.1%)	82 (50.3%)		19 (54.3%)	53 (54.6%)	
**Ann Arbor Stage, *n* (%)**							
IA-IB-IIA	80 (37.7%)	14 (28.6%)	66 (40.5%)	0.66	NA	NA	
IIB	57 (26.9%)	16 (32.7%)	41 (25.2%)		16 (45.7%)	41 (42.3%)	0.65
III	43 (20.3%)	10 (20.4%)	33 (20.2%)		10 (28.6%)	33 (34.0%)	
IV	32 (15.1%)	9 (18.4%)	23 (14.1%)		9 (25.7%)	23 (23.7%)	
**B-symptoms, *n* (%)**							
No	96 (45.3%)	18 (36.7%)	78 (47.9%)	0.19	4 (11.4%)	12 (12.4%)	0.98
Yes	116 (54.7%)	31 (63.3%)	85 (52.1%)		31 (88.6%)	85 (87.6%)	
**WBC** median (range)	9.86 (1.12–26.00)	11.78 (3.10–26.00)	9.60 (1.12–24.34)	0.24	11.00 (3.10–26.00)	10.50 (1.67–24.34)	0.17
**ANC** median (range)	7.24 (0.79–22.20)	8.37 (1.60–2.22)	6.92 (0.79–20.56)	0.19	8.87 (1.60–22.20)	7.89 (0.81–20.56)	0.13
**ALC** median (range)	1.46 (0.19–3.95)	1.60 (0.19–3.30)	1.45 (0.19–3.95)	0.92	1.60 (0.19–3.23)	1.39 (0.24–3.95)	0.92
**AMC** median (range)	0.62 (0.17–1.86)	0.68 (0.17–1.80)	0.59 (0.26–1.86)	0.89	0.74 (0.17–1.81)	0.66 (0.25–1.85)	0.88
**NLR** median (range)	4.79 (0.60–64.7)	7.69 (1.50–64.7)	6.16 (0.60–53.4)	0.09	6.20 (1.51–64.7)	5.50 (0.61–48.8)	0.11
**Large nodal mass, *n* (%)**							
≤7 cm	148 (69.8%)	34 (69.4%)	114 (69.9%)	0.98	25 (71.4%)	63 (64.9%)	0.53
>7 cm	64 (30.2%)	15 (30.6%)	49 (30.1%)		10 (28.6%)	34 (35.1%)	
**Extranodal sites, *n* (%)**							
Absent	159 (75.0%)	34 (69.4%)	125 (76.7%)	0.35	20 (57.1%)	66 (68.0%)	0.31
Present	53 (25.0%)	15 (30.6%)	38 (23.3%)		15 (42.9%)	31 (32.0%)	
**IPS score, *n* (%)**							
<3	158 (74.5%)	35 (71.4%)	123 (75.5%)	0.59	24 (68.6%)	63 (64.9%)	0.83
≥3	54 (25.5%)	14 (28.6%)	40 (24.5%)		11 (31.4%)	34 (35.1%)	
**PET-2 status, *n* (%)**							
Negative	179 (84.4%)	40 (81.6%)	139 (85.3%)	0.51	26 (74.3%)	78 (80.4%)	0.98
Positive	33 (15.6%)	9 (18.4%)	24 (14.7%)		9 (25.7%)	19 (19.6%)	
**Response, *n* (%)**							
cCR	188 (88.7%)	36 (73.5%)	152 (93.3%)	*0.0004*	24 (68.6%)	89 (91.8%)	*0.002*
relapse/refractoriness	24 (11.3%)	13 (26.5%)	11 (6.7%)		11 (31.4%)	8 (8.2%)	

**Table 2 cancers-16-00826-t002:** Evaluation of the strength of the association between baseline levels of IgA, IgG, and IgM and established prognostic factors in HL (Fisher exact test).

Clinical Variable	Low IgM		Low IgG		Low IgA	
	Chi-Squared	*p*	Chi-Squared	*p*	Chi-Squared	*p*
Age ≥ 45-years-old	0.005	0.94	0.15	0.69	1.28	0.52
Male gender	0.34	0.56	0.34	0.55	2.31	0.31
Albumin < 4.0 g/dL	3.66	0.06	0.19	0.65	0.55	0.75
ALC < 600 cells/μL	0.11	0.73	0.38	0.53	1.99	0.36
NLR ≥ 6	0.33	0.59	0.21	0.66	0.94	0.60
WBC ≥ 15,000 cells/μL	0.16	0.69	0.18	0.66	3.26	0.19
Large nodal mass > 7 cm	0.005	0.94	0.007	0.93	0.38	0.83
Presence of extranodal sites	1.07	0.31	0.44	0.50	2.07	0.35
Hemoglobin < 10.5 g/dL	1.19	0.27	0.25	0.61	1.46	0.48
LDH > 2 UNL	0.05	0.82	0.18	0.67	0.38	0.82
IPS ≥ 3	0.32	0.57	0.19	0.65	4.44	0.10
Positive PET-2 status	0.38	0.53	0.007	0.93	0.96	0.62

**Table 3 cancers-16-00826-t003:** Univariate analysis of PFS of all patients included in the study (*n* = 212).

Clinical Variable	N	5-Year PFS	*p*-Value	HR	95% CI
**Age**					
≤45 years	148	74.7%	0.85	0.95	0.53–1.69
>45 years	64	75.7%			
**Gender**					
Male	109	71.7%	0.14	0.67	0.39–1.13
Female	103	78.4%			
**WBC**					
<15.000 cells μ/L	177	78.6%	*0.005*	0.44	0.21–0.94
≥15.00 cells μ/L	35	57.1%			
**ALC**					
<600 cells μ/L	15	58.3%	0.15	0.54	0.18–1.62
≥600 cells μ/L	197	76.2%			
**Bulky disease**					
Absent	148	79.3%	*0.02*	0.53	0.29–0.97
Present	64	64.9%			
**Extranodal disease**					
Absent	159	78.8%	*0.03*	0.55	0.29–1.03
Present	53	63.5%			
**IPS**					
<3	158	78.7%	*0.01*	0.51	0.27–0.96
≥3	54	63.8%			
**NLR**					
<6	127	80.7%	*0.008*	2.09	1.21–3.60
≥6	85	66.5%			
**IgM**					
≤50 mg/dL	49	54.9%	*<0.0001*	3.33	1.69–6.53
>50 mg/dL	163	81.1%			
**PET2-status**					
Negative	179	83.4%	*<0.0001*	0.14	0.06–0.34
Positive	33	29.5%			

**Table 4 cancers-16-00826-t004:** Univariate analysis of PFS in advanced-stage patients (*n* = 132).

Clinical Variable	N	5-Year PFS	*p*-Value	HR	95% CI
**Age**					
≤45 years	90	70.7%	*0.91*	0.96	0.49–1.88
>45 years	42	55.2%			
**Gender**					
Male	72	77.2%	*0.06*	NA	NA
Female	60	60.9%			
**WBC**					
<15.000 cells μ/L	103	75.0%	0.012	2.16	0.97–4.83
≥15.00 cells μ/L	29	55.2%			
**ALC**					
<600 cells μ/L	10	33.3%	*0.007*	3.09	0.78–12.16
≥600 cells μ/L	122	73.2%			
**Bulky disease**					
Absent	88	74.7%	*0.02*	1.99	1.01–3.96
Present	44	56.1%			
**Extranodal disease**					
Absent	86	74.8%	*0.04*	1.89	0.96–3.71
Present	46	57.1%			
**IPS**					
<3	87	75.3%	*0.01*	2.16	1.07–4.33
≥3	45	54.8%			
**NLR**					
<6	70	75.8%	*0.002*	2.09	1.10–3.95
≥6	62	60.1%			
**IgM**					
≤50 mg/dL	35	49.9%	*0.0002*	3.13	1.46–6.72
>50 mg/dL	97	75.5%			
**PET2-status**					
Negative	104	80.5%	*<0.0001*	6.42	2.60–15.82
Positive	28	25.2%			

**Table 5 cancers-16-00826-t005:** Multivariate analysis of PFS in patients included in the study, according to clinical predictors, including the PET-2 status.

Clinical Predictor	All Patients				Advanced-Stage Patients			
	Clinical Variables Available at Baseline		All Clinical Variables (Including PET2 Status)		Clinical Variables Available at Baseline		All Clinical Variables (Including PET2 Status)	
	HR (95%CI)	*p*	HR (95%CI)	*p*	HR (95% CI)	*p*	HR (95%CI)	*p*
**NLR ≥ 6**	1.52 (0.86–2.69)	0.15	1.43 (0.82–2.48)	0.21	1.81 (0.92–3.56)	0.08	1.83 (0.93–3.59)	0.08
**WBC** **≥15,000 cells/μL**	1.97 (1.04–3.71)	*0.04*	1.52 (0.81–2.87)	0.19	NA	NA	NA	NA
**Positive PET-2**	NA	NA	6.17 (3.42–11.12)	*<0.0001*	NA	NA	6.39 (3.21–12.73)	*<0.0001*
**Large nodal mass** **> 7 cm**	1.55 (0.89–2.71)	0.12	1.33 (0.76–2.32)	0.32	2.00 (1.04–3.84)	*0.036*	1.69 (0.87–3.25)	0.12
**Presence of** **extranodal sites**	1.69 (0.96–2.95)	0.07	1.36 (0.77–2.42)	0.29	1.59 (0.78–3.24)	0.20	1.51 (0.73–3.09)	0.26
**IPS ≥ 3**	NA	NA	NA	NA	2.05 (0.99–4.22)	0.051	1.80 (0.86–3.76)	0.12
**IgM < 50 mg/dL**	3.45 (2.02–5.91)	*<0.0001*	4.02 (2.32–6.97)	*<0.0001*	3.43 (1.81–6.52)	*0.0002*	4.32 (2.23–8.38)	*<0.0001*

## Data Availability

The data presented in this study are available on request from the corresponding author. The data are not publicly available due to privacy and ethical restrictions.

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
