# Peer review of "Baseline IgM Amounts Can Identify Patients with Poor Outcomes: Results from a Real-Life Single-Center Study on Classical Hodgkin Lymphoma"

_cancers, 2024, doi:10.3390/cancers16040826_

Round 1
Reviewer 1 Report
Comments and Suggestions for Authors
1- Line 17- "has "a" prognostic..... Line 24- PET-2, "and" guaranteeing ....
2- Line 49 "performed early "after the rather than early performed. Line 63- Increasing "amount" of evidence, rather than number of evidence.
3- Line 86- Immunoglobulins? Which immunoglobulins, clarify.
4- Start sentence with "Although" never tested in......
5- No problem with Table 1. Line 175 were allocated to "an" escalated...
6- Figure 2 and 4 are good to predict treatment failure, and negative patients.
7- In all, I believe this is a good study to understand the various factors that can drive disease to become more aggressive.
Reviewer 2 Report
Comments and Suggestions for Authors
The authors have expertly performed a study on a large number of HL patients, in an attempt to identify factors that could be used to predict outcome and guide therapy. Their data support a remarkable correlation of the baseline IgM level with PFS, for both Advanced-Stage and Early-Stage patients. The manuscript could be improved by addressing the following points:
1) Figure 1 is very useful for comprehending the entire dataset, with respect to stage and results of PET2&3. It might be similarly useful to make a similar figure in which the initial branch points for each group (Advanced-Stage and Early-Stage) are based on the IgM level (< 50 vs. >50), and then the PET-2 and PET-3 results are shown graphically, such as by bars with the proportion of positive and negative results.
2) In this analysis, the baseline IgM level is dichotomized at the level of 50 mg/dL. While this simplifies analysis and comprehension, there is potential for error in choosing this particular breakpoint, and inevitably a loss of information about what is actually a continuous variable. What can the authors tell us about how this particular level was chosen? E.g., does a histogram of the actual values show a trough at this point? And how does IgM level perform as a continuous variable in a Cox analysis?
3) Based on the way in which the analysis of this dataset was performed, it is possible that the implication of IgM level as a significant predictive variable is simply the result of overfitting. This seems unlikely, especially since IgM level performed well in both Advanced-Stage and Early-Stage subsets of the dataset. However, the authors should address this possibility, preferably by additional analysis (test vs. training, LOOCV, etc.).
4) The Discussion should mention the recent publication PMID: 38081297, which shows the predictive power of ctDNA of HL patients at baseline and after 2 cycles.
